# Ionosonde Total Electron Content Evaluation Using IGS Data

Telmo dos Santos Klipp[1], Adriano Petry[1], Jonas Rodrigues de Souza[2], Eurico Rodrigues de Paula[2], Gabriel Sandim Falcão[1], and Haroldo Fraga de Campos Velho[2]

[1]National Institute for Space Research, Southern Regional Space Research Center, Av Roraima, campus UFSM, prédio do INPE/CRS, sala 2023, PO box 5021, zipcode 97105-970, Santa Maria, RS, Brazil, Phone: +55 55 33012012.
[2]National Institute for Space Research, Av. dos Astronautas, 1758 - Jardim da Granja, São José dos Campos/SP - CEP 12227-010 - Brasil.

**Correspondence:** Telmo dos Santos Klipp (telmo.klipp@gmail.com)

**Abstract.** In this work, a period of two years (2016-2017) of Ionospheric Total Electron Content (ITEC) from ionosondes operating in Brazil is compared to the International GNSS Service (IGS) vertical Total Electron Content (vTEC) data. Sounding instruments from National Institute for Space Research (INPE) provided the ionograms used, which were filtered based on confidence score (CS) and C-level flags evaluation. Differences between vTEC from IGS maps and ionosonde TEC were accumulated in terms of root mean squared error (RMSE). As expected, it was noticed the ITEC values provided by ionosondes are systematically underestimated, which is attributed to a limitation in the electron density modeling for the ionogram topside that considers a fixed scale height, which makes density values decay too rapidly above ∼800 km, while IGS takes in account electron density from GNSS stations up to the satellite network orbits. The topside density profiles covering the plasmasphere were re-modeled using two different approaches: an optimization of the adapted $\alpha$-Chapman exponential decay that includes a transition function between the F2 layer and plasmasphere, and a corrected version of the NeQuick topside formulation. The electron density integration height was extended to 20,000 km to compute TEC. Chapman parameters for the F2 layer were extracted from each ionogram, and plasmaspheric scale height was set to 10,000 km. A criterion to optimize the proportionality coefficient used to calculate the plasmaspheric basis density was introduced in this work. The NeQuick variable scale height was calculated using empirical parameters determined with data from Swarm satellites. The mean RMSE for the whole period using adapted $\alpha$-Chapman optimization reached a minimum of 5.32 TECU, that is 23% lower than initial ITEC errors, while for NeQuick topside formulation the error was reduced 27%.

## 1 Introduction

The understanding of ionospheric behaviour provides important information about the space weather. In addition, the electron content affects group and phase delays of radio waves passing through ionosphere and impacts, among other, global navigation satellite systems (GNSS). Different instruments are capable of evaluating electron density in ionosphere, and validations among different sources of data can lead to interesting conclusions. While ionosonde instruments provide "ground truth" measures for the bottom side of ionospheric profile and estimate the topside using exponential decay function, ground GNSS stations receiving radio signals from orbiting satellites can provide large scale details of the entire ionosphere structure and even

plasmasphere (Huang and Reinisch, 2001; Reinisch and Huang, 2001; Jakowski, 2005; Reinisch and Galkin, 2011; Jin and Jin, 2011). The analysis proposed in this work is based on comparisons between TEC estimated using density profiles derived from ionograms, and vertical Total Electron Content (vTEC) from the International GNSS Service (IGS). While IGS has its own intrinsic quality control through a ranking system (Hernández-Pajares et al., 2009), ionosonde data is evaluated by its auto-scaling system, and quality scores are assigned to each ionogram. The study was performed in Brazilian region for a 2-year period (2016-2017), where ionosonde data from National Institute for Space Research (INPE) were available. Indeed, the plasmaspheric electron density has been considered using 2 different models: an adapted $\alpha$-Chapman function (Jakowski, 2005) with a simple optimization, and a corrected version of the NeQuick topside formulation (Pezzopane and Pignalberi, 2019).

## 1.1 IGS vTEC maps

IGS vTEC maps are considered to be a reliable ionospheric information product, which was achieved from integrating scientific community efforts (Hernández-Pajares et al., 2009). Such maps are generated by a combination of data from different research institutions within a method that is based on ranking different vTEC maps to compose the final product (Hernández-Pajares et al., 2009). The process begins with raw data from the GNSS ground network being acquired and sent to ionospheric associate analysis centers (IAACs) so the vTEC maps can be generated using the ionosphere map exchange (IONEX) format (Schaer et al., 1998). To achieve a high level of quality, these vTEC maps are evaluated, and its ability to reproduce corresponding slant TEC (STEC) maps is checked. Next, a combination process takes place using a weighted mean of the available vTEC maps. The final step before making the maps available for access on the IGS server is a validation process. It compares the vTEC maps to an independent source: dual frequency altimeters on board TOPEX, JASON and ENVISAT satellites.

## 1.2 Ionosonde data

An ionosonde measures the returning echoes of pulse signals at a fixed location to estimate ionospheric characteristics, and the ionogram trace can be processed to result in a vertical electron density profile. The bottom side profile starts with measures at $\sim$90 km up to the peak of the F2 layer ($f_0F2$), around 350 km. The ionosphere topside profile, instead, is modeled using an exponential decay function. The integration of electron density in height produces an estimate for the TEC value.

Ionograms can be interpreted either manually by expert or automatically using software. The autoscaling ionosonde data availability, rather than manual scaling, contributes to meet practical applications (Jiang et al., 2015). Different systems were created and concentrated efforts have been applied to improve autoscaling (Reinisch and Xueqin, 1983; Scotto and Pezzopane, 2002, 2007; Reinisch et al., 2005). Also, a standard archiving output (SAO) format was created by initiative of the Ionospheric Informatics Working Group (IIWG) to store and disseminate auto-scaled data. Initially, SAO format considered only ionograms scaled by automatic real-time ionogram scaler with true height (ARTIST), however, it evolved to hold scaled data from others sounder systems (Galkin, 2006).

### 1.2.1 Ionogram quality

Several attempts had been made to verify ionogram auto-scaling system quality (Reinisch et al., 2005; Enell et al., 2016; Pezzopane et al., 2017). Early comparisons between manual and automatic scaled ionospheric parameters revealed limitations on ARTIST system performance due to the absence of quality metrics (Gilbert and Smith, 1988). Recent versions of ARTIST system improved quality proof methods, enhancing their results (Bamford et al., 2008; Galkin and Reinisch, 2008) and facing problems related to autoscaling (Pezzopane and Scotto, 2007; Stankov et al., 2012).

Ionosonde data scaled by ARTIST 5 have a quality metric called confidence score (CS). Such metric is based on quality criteria supported by concepts of ionogram interpretation and algorithms that specify the uncertainty and confidence of scaled results (Galkin et al., 2013). The CS metric includes quality checking solutions introduced by confidence calculation schemes developed since late 1980s: the auto-scaling confidence level (ACL) quality flag, the two-digit confidence level (C-Level) and the QualScan quality control (McNamara, 2006; Galkin et al., 2013). The estimation of confidence score occurs during ionogram processing. Ionogram interpretation criteria consider not only analysis of extracted trace shapes, but ionospheric conditions to compute per-point-error reduction. The CS starts with a value of 100. If an interpretation criterion is found, its per-point-error value is subtracted from CS. To be considered acceptable for further use, auto-scaling records need to reach a CS above a predefined threshold value, which is generally 40 (Galkin et al., 2013).

The SAO format version 4 does not have the CS on its specifications, but have the C-Level representation. The two digits range goes from 11 (highest confidence) to 55 (lowest confidence). The CS produced by ARTIST 5 can be converted to C-Level representation using the Table 1 (Galkin et al., 2013).

## 2 Methodology

Ionosonde data was obtained from INPE database using files in SAO format (version 4) and scaled by auto-scaling system ARTIST (version 5). Data from up to 5 instruments (see Fig. 1) were available at the same time for the period considered (2016-2017). Although ionosondes can generate ionograms in less than ten minutes interval, an 1 hour interval between soundings was considered in this work, except for the comparisons with IGS data. In that case, 2 hours interval is used to match IGS data availability.

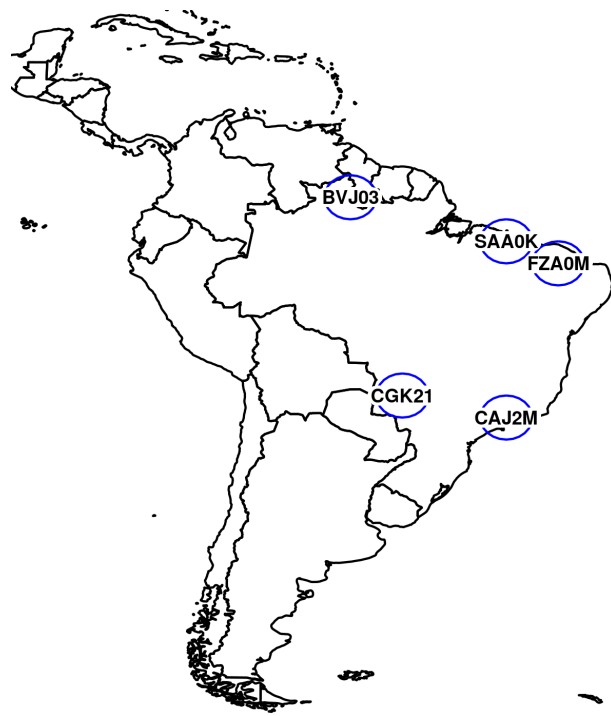

**Figure 1.** Location of available ionosonde data during 2016-2017.

C-Level values were extracted from all ionograms, and despite auto-scaled ionosonde data with CS above 40 can be considered acceptable (Galkin et al., 2013), we chose to use only those achieving a CS above 60, corresponding to C-level 11 and 22. Figure 2 shows the total number of C-Level flags occurrences for the available data. Figure 3 shows the same distribution, however, considering each ionosonde station separately. It can be seen from Figs. 2 and 3 that the majority of C-Level flags occurrences are in classification levels 11 or 22. Also, more than half of C-Level flags achieve CS above 80 (C-level 11).

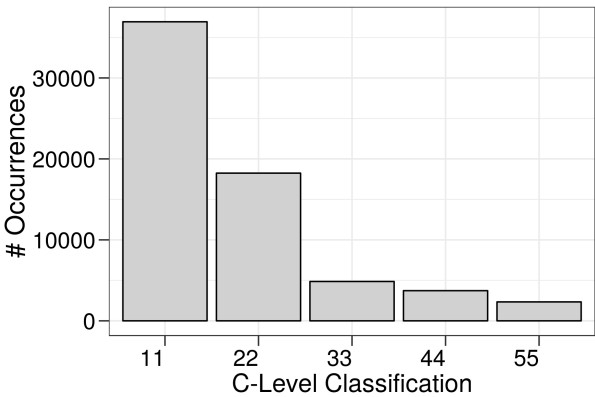

**Figure 2.** Distribution of C-Level flags for ionograms during 2016-2017.

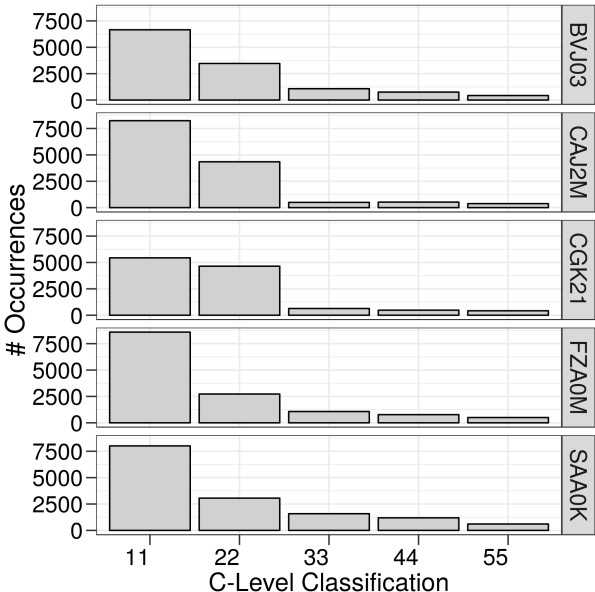

**Figure 3.** The same as Fig. 2 but for each ionosonde station separately.

Considering the daily variation in ionosphere electron density, it would be interesting to analyse also the ionosondes' data quality variation within day hours taking into account all available data. It can be seen in Fig. 4 the increase in the occurrence of level 11 after sunset while the level 22 decreases. The aggregation of C-level flags 11 and 22 ionograms (used in this work) provides ionosonde data with over 1000 samples even for the period with low occurrences - see red curve in Fig. 4.

5    TEC values from IGS maps and ionosondes were compared at the same date/time using the closest geographic correspondence as shown in Table 2, considering IGS data grid (5° in longitude per 2.5° in latitude, every 2 hours). The analysis is

mainly based on the accumulation of TEC differences by applying the root mean squared error (RMSE) as defined in Eq. (1) (Chai and Draxler, 2014):

$$RMSE = \sqrt{\frac{1}{n}\sum_{i=1}^{n}e_i^2} \qquad (1)$$

where error $e_i$ are the differences (with $i=1,2,...,n$) between TEC values from IGS and ionosonde, and $n$ is the total number of values considered.

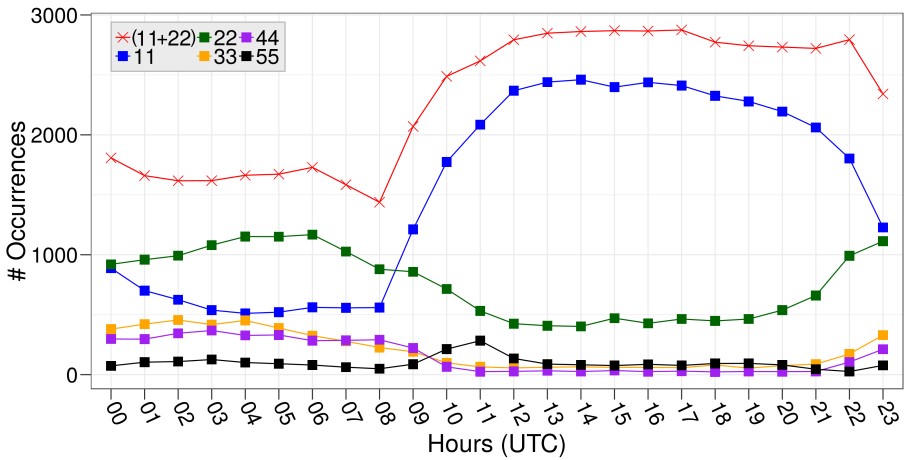

**Figure 4.** Hourly distribution of C-level flags.

## 3 Experiments and results

Ionospheric Total Electron Content (ITEC) daily variability for each ionosonde C-level flag is shown in Fig. 5, and for each ionosonde station separately, considering only C-level flags 11 and 22, is shown in Fig. 6. Since the daily mean ITEC values from flags 11 and 22 follow the IGS vTEC data variation, they are coherent. On the other hand, it can be seen the vTEC values are consistently higher than ITEC for the whole period and for every ionosonde. It is also noticeable in Fig. 5, a noisy and incoherent TEC variation for flags greater than 22 in both vTEC and ITEC. Obviously, since the results for flags greater than 22 have low confidence, they may have errors, but this reason can not be used to explain the noise in vTEC values. The data representative for higher flags is low, i.e., there is reduced number of points and heterogeneous distribution during day and nighttime. Such unbalanced distribution can produce daily mean TEC representing only day or nighttime, that during consecutive days lead to noisy curves.

In Fig. 6 we can observe that some ionosondes presented lack of data for few days or even entire months. The seasonal variation in ITEC was similar for all stations. During autumn and winter seasons in southern hemisphere we can notice a decrease in ITEC values for both years evaluated.

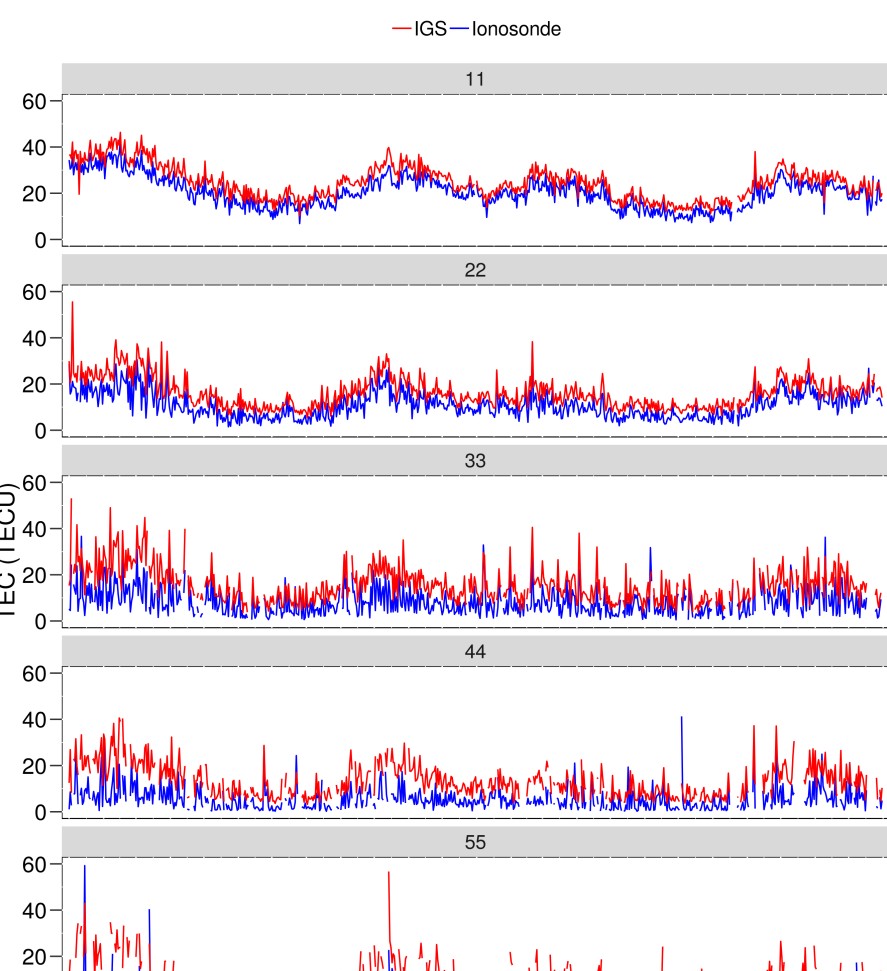

**Figure 5.** vTEC and ITEC daily mean variation for the period under analysis considering the mean value for each C-level flag classification.

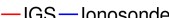

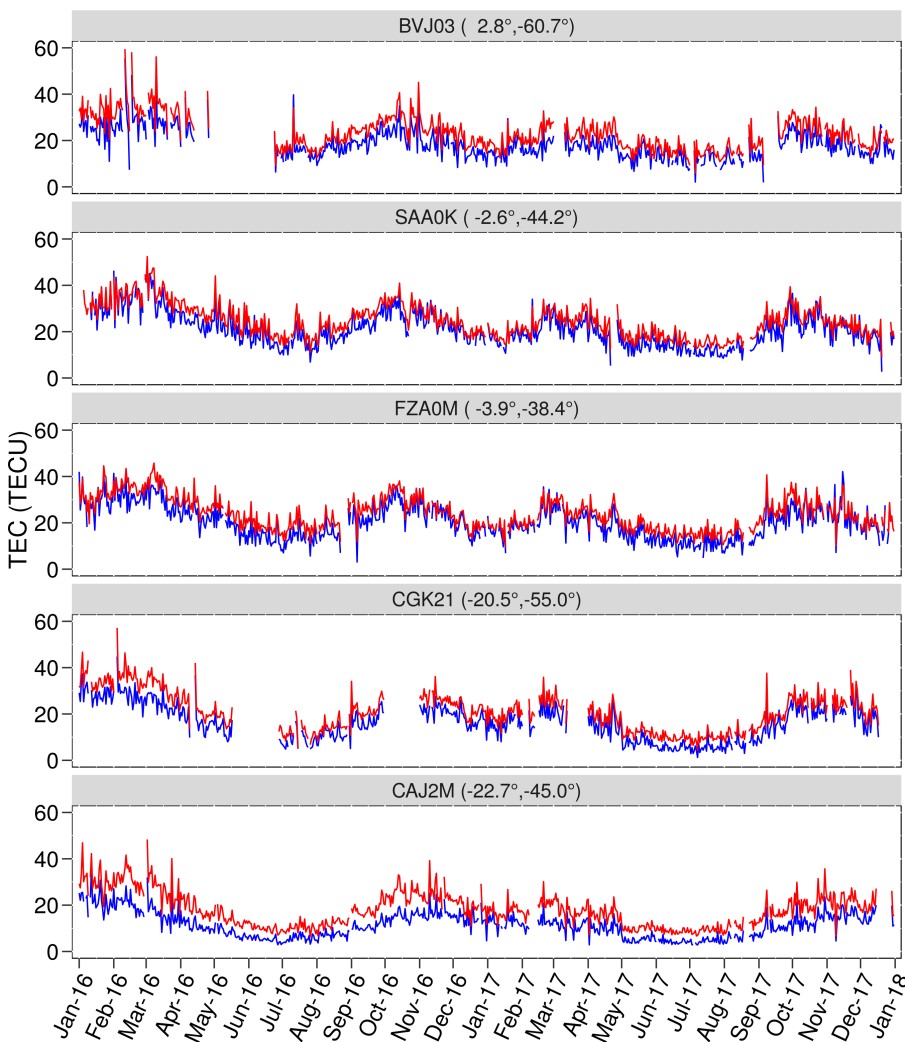

**Figure 6.** The same as Fig. 5 but considering only C-level flags 11 and 22 and for each ionosonde station separately.

All panels in Figure 7 present daily mean values, considering the density profiles of all ionosondes. In (a) it is shown the RMSE when comparing ITEC and IGS vTEC. The seasonal variability in TEC differences seems highly correlated to ionization distribution along the analysed period. Figure 7 (b) shows the peak of plasma frequency ($f_0F2$), and we can observe the periods of high $f_0F2$ values correspond to high RMSE. The maximum altitude used for electron density integration in ionosondes (Fig. 7 (c)) does not change significantly, rarely reaching 900 km. Figure 7 (d) shows the plasma frequency at the maximum altitude of density profile, which indicates the level from where it is necessary an extension of ionosphere structure evaluation to higher altitudes. Considering the fixed scale height used in digisonde topside profile modeling, such contribution

has not been included in the ITEC calculation, since values of electron density decay too rapidly above ∼800 km, and the simple extension of maximum integration altitude is insufficient for proper comparisons to IGS vTEC. This is the main reason why the ITEC values from ionosondes are underestimated when compared to IGS values. It is well known that vTEC values from the IGS data represent the integrated electron density along the signal path between the receiver and the satellite altitude (∼20,000 km). Thus, this analysis is in agreement with what is shown in Fig. 5 and 6.

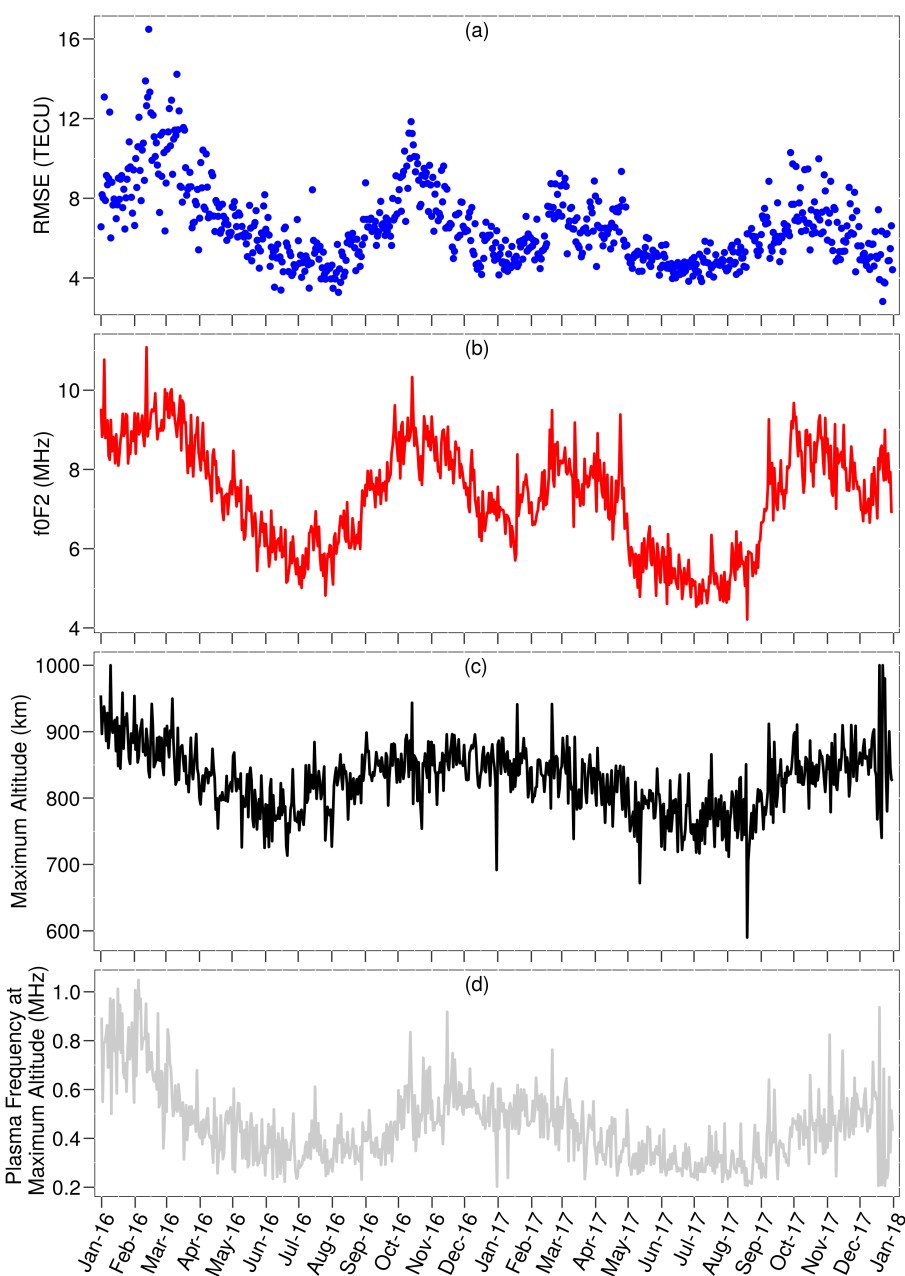

**Figure 7.** Daily mean, considering all ionosondes: (a) ITEC and IGS vTEC differences in terms of RMSE; (b) F2 layer critical frequency; (c) maximum altitude of density profile and (d) plasma frequency at maximum altitude.

Different analytical functions have been used to model the topside ionospheric density profile (e.g. exponential, Epstein, Chapman) (Nsumei et al., 2012; Pignalberi et al., 2018a; Reinisch et al., 2007). These functions and their variations may adopt

fixed or variable scale height. In this work, the ionogram topside profiles were re-modeled using two different approaches that consider fixed and variable scale height: an adapted $\alpha$-Chapman exponential decay (Jakowski, 2005) that includes an optimized transition function between the F2 layer and plasmasphere, and the NeQuick topside formulation with modeled scale height as a function of a corrected version of the empirical parameter $H_0$ (Pezzopane and Pignalberi, 2019).

5   ## 3.1   Adapted $\alpha$-Chapman

The adapted $\alpha$-Chapman introduced by Jakowski (2005), defines the topside profile $N_T$ as

$$N_T(h) = NmF2 \cdot exp(\frac{1}{2}(1 - z - e^{-z})) + N_{P0} \cdot exp(\frac{-h}{H_p}), \quad where \quad z = \frac{h - hmF2}{H_T} \tag{2}$$

The ionograms provided F2 scale height $H_T$, the electron peak density $NmF2$ that can be derived from measured critical frequency $f_0F2$ using $NmF2 = (1/80.6) \cdot (f_0F2)^2$, and peak height $hmF2$. According to Jakowski (2005), the plasmaspheric
10   scale height $H_p$ can be defined as 10,000 km and the plasmaspheric basis density $N_{P0}$ is assumed to be proportional to $NmF2$, i.e., $N_{P0} = K \cdot NmF2$. Using the topside reconstruction of density profile shown above, the maximum integration height used to estimate ionosonde TEC values was defined as 20,000 km, corresponding to an approximation for the satellites orbit.

In this work, different values for the proportionality coefficient $K$ are examined, and the optimal factor that minimizes the global RMSE is used. Fig. 8 shows the ionosonde TEC differences to IGS vTEC in terms of mean RMSE in the whole period,
15   considering all ionosondes. When $K$ is set to zero, Eq. (2) is reduced to regular $\alpha$-Chapman decay, and plasmaspheric slowly decaying exponential term is ignored. As $K$ increases, the underestimated ionosonde TEC values move closer to IGS, hence reducing RMSE. However, after an optimal $K$, in this experiment equals to $1/175$, the plasmaspheric contribution is exceeded, increasing again RMSE.

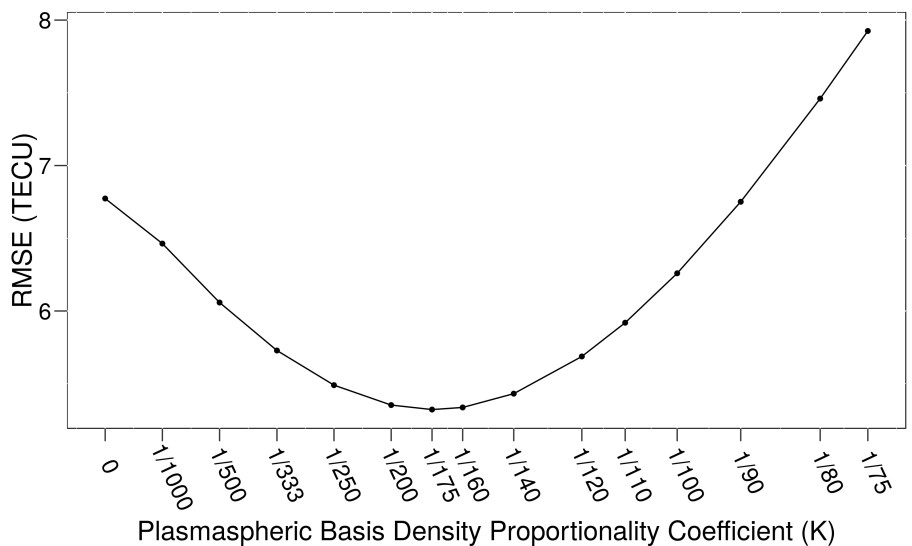

**Figure 8.** Variation of total RMSE with plasmaspheric basis density proportionality coefficient.

### 3.2 NeQuick topside formulation

The NeQuick topside analytical formulation (Nava et al., 2008; Pezzopane and Pignalberi, 2019) is based on a semi-Epstein layer describing the topside electron density profile $N_T$ as

$$N_T(h) = 4 \cdot NmF2 \cdot \frac{exp(z)}{(1 + exp(z))^2}, \quad where \quad z = \frac{h - hmF2}{H_T} \tag{3}$$

5     In this approach, the modeled scale height $H_T$ is dependent on height $h$, $hmF2$, and also on the empirical parameter $H_0$:

$$H_T(h) = H_0 \cdot \left[ 1 + \frac{100 \cdot 0.125 \cdot (h - hmF2)}{100 \cdot H_0 + 0.125 \cdot (h - hmF2)} \right] \tag{4}$$

$H_0$ can be calculated using $NmF2$, $f_0F2$, the propagation factor ($M(3000)F2$), $hmF2$ and the smoothed sunspot number ($R_{12}$) as presented by Nava et al. (2008). Pezzopane and Pignalberi (2019) proposed a new formulation for $H_0$ based on electron density measurements made by the Swarm satellite constellation. The formulation is

$$
10 \quad H_0 = 
\begin{cases}
H_{0,AC} + (H_{0,B} - H_{0,AC}) \cdot \frac{h - hmF2}{600}, & for \quad hmF2 \leq h < hmF2 + 600 \\
H_{0,B}, & for \quad h \geq hmF2 + 600,
\end{cases}
\tag{5}
$$

where 2 two-dimensional grids provide the values of $H_{0,AC}$ and $H_{0,B}$ as a function of $f_0F2$ and $hmF2$. Specifically, the grids have been calculated as the median values obtained by using the NeQuick topside formulation, IRI UP modeled values

(Pignalberi et al., 2018b, c), Swarm A and C (for $H_{0,AC}$) or Swarm B (for $H_{0,B}$) electron density measurements (Pezzopane and Pignalberi, 2019). In our experiments, $hmF2$ and $f_0F2$ were obtained from the post-processed ionograms and $H_{0,AC}$ and $H_{0,B}$ grids were gently provided by Dr. Michael Pezzopane and Dr. Alessio Pignalberi from the Istituto Nazionale di Geofisica e Vulcanologia, Italy.

5 ### 3.3 Comparative evaluation

Fig. 9 presents a comparison, considering all ionosondes, among the daily mean ITEC (in blue), IGS vTEC (in red), density profile integration up to 20,000 km with topside reconstruction using adapted $\alpha$-Chapman function (in orange) and NeQuick formulation (in green). The results are for the years 2016 (top panel) and 2017 (bottom panel). All calculated values follow the reference, i.e., the IGS vTEC variations showing a semi-annual dependency with a minimum in June solstice as well as the

10 day-to-day variability. Indeed, the TEC values obtained with the optimizition of an adapted $\alpha$-Chapman are very similar to the ones from NeQuick topside procedure and both are much closer to IGS vTEC than ITEC values.

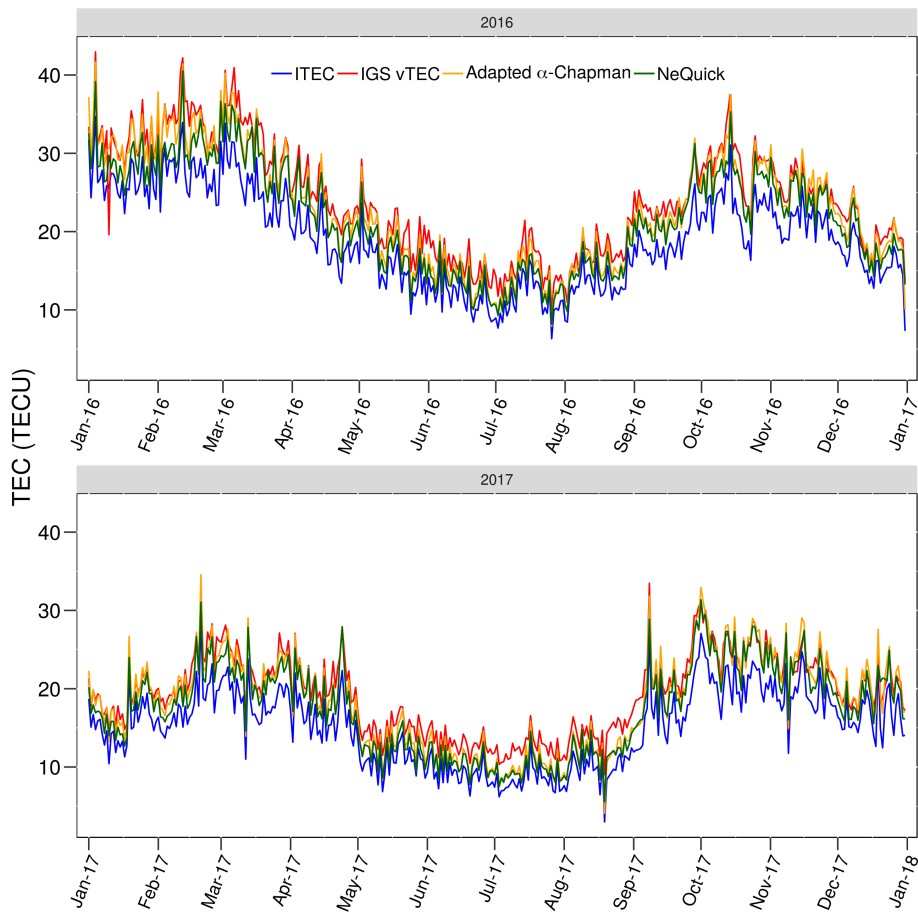

**Figure 9.** Daily mean, considering all ionosondes, of ITEC (blue), IGS vTEC (red) and density profile integration up to 20,000 km with topside reconstruction using adapted $\alpha$-Chapman (orange) and NeQuick formulation (green).

To identify the best methodology to estimate the plasmaspheric TEC, the daily mean RMSE variation shown in Fig. 10 can be assessed. The differences to IGS vTEC using adapted $\alpha$-Chapman (orange triangle) and NeQuick (green cross) approaches can be compared to the ITEC errors (blue circle) as shown in Fig. 7 (a). In general, the RMSE values calculated with adapted $\alpha$-Chapman and NeQuick are similar. However, the lowest RMSE values along the two years 2016 and 2017 belong to the
5   NeQuick criterion, as shown in Fig. 10. In addition, the mean RMSE for the whole period, using adapted $\alpha$-Chapman reconstruction with the proportionality coefficient optimization, has a minimum value of 5.32 TECU, while using NeQuick topside reconstruction the error is 5.05 TECU.

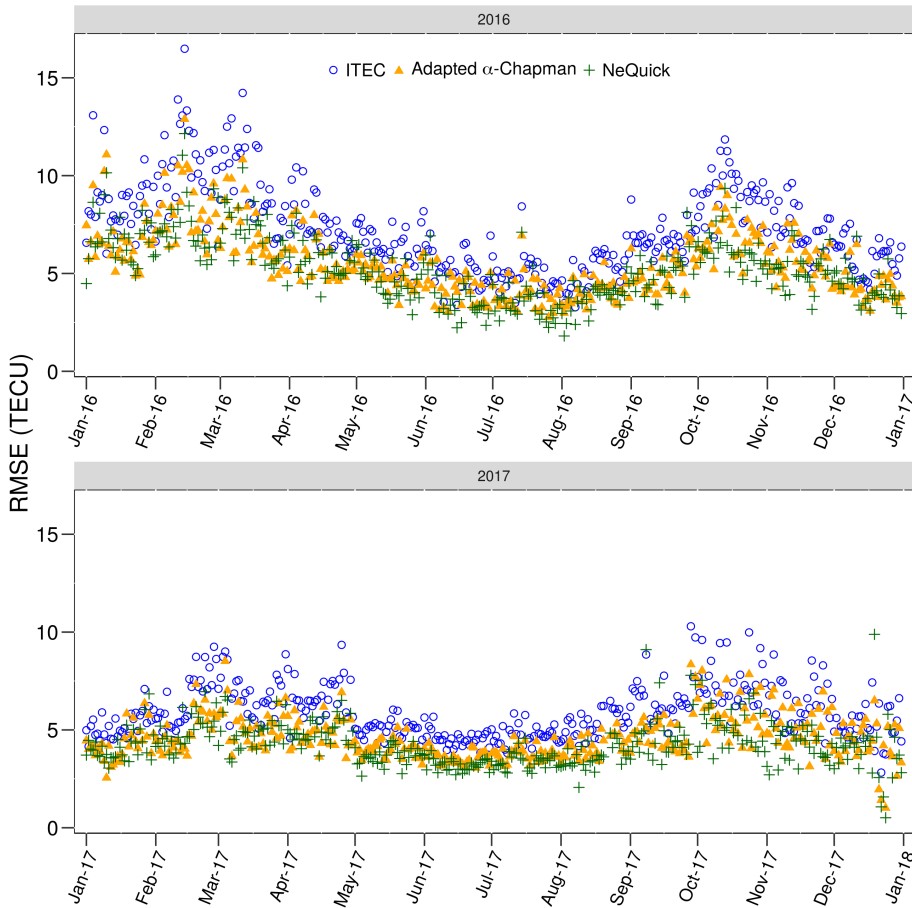

**Figure 10.** Daily mean RMSE variation, considering all ionosondes, for ITEC (blue circle), adapted $\alpha$-Chapman (orange triangle) and NeQuick topside formulation (green cross).

## 4 Conclusions

This paper presented a 2-year period validation of ionosonde data, using IGS vTEC as reference. Ionogram electron density profiles were first selected based on the Confidence Score, and then integrated in height. As expected, ITEC values were systematically underestimated, what is consistent to ionospheric topside modeling limitation that uses a fixed scale height,

5 which almost neglect plasma above $\sim$800 km, while IGS data considers electron densities from the GNSS stations up to the satellite. This claim was supported by the examination of Figs. 5, 6 and 7. The ionogram topside profiles were re-modeled using two different approaches: optimization of an adapted $\alpha$-Chapman exponential decay and the NeQuick topside formulation, based on a semi-Epstein layer with modeled scale height as a function of a corrected version of the $H_0$ empirical parameter. The electron density integration height was extended to an approximation of satellite orbits. Hence, as expected, for both

10 topside reconstructions the plasmaspheric ionization contribution brought ionosonde TEC values closer to IGS observations.

In our experiments, the improvement was significant to determine TEC using ionosonde data, as shown in Figs. 9 and 10. Although both procedures for calculating plasmaspheric TEC yield similar results, the NeQuick criterion shows lower RMSE values, as we can clearly see in Fig. 10.

*Competing interests.* The authors declare that they have no conflict of interest.

*Acknowledgements.* T. S. Klipp would like to acknowledge Conselho Nacional de Desenvolvimento Científico e Tecnológico (CNPq, Brazil) for Programa de Capacitação Institucional PCI-DC research sponsorship under the grant 300487/2019-3.

G. S. Falcão would like to acknowledge CNPq for PIBIC sponsorship.

J. R. Souza and E. R. Paula would like to thank the CNPq (under the respective grants 307181/2018-9 and 310802/2015-6) for research productivity sponsorship and the INCT GNSS-NavAer supported by CNPq (465648/2014-2), FAPESP (2017/50115-0) and CAPES (88887.137186/2017-00).

H. F. C. Velho would like to acknowledge CNPq for research productivity sponsorship.

The authors would like to acknowledge the Crustal Dynamics Data Information System (CDDIS), NASA Goddard Space Flight Center, Greenbelt, MD, USA for providing online access to IGS vTEC data at ftp://cddis.gsfc.nasa.gov/gnss/products/ionex/igsg.

The authors thank Dr. Michael Pezzopane and Dr. Alessio Pignalberi from the Istituto Nazionale di Geofisica e Vulcanologia, Italy, for providing deeper understanding about implementation of the NeQuick topside formulation, and access to median values of $H_{0,AC}$ and $H_{0,B}$ as a function of $f_0F2$ and $hmF2$.

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

**Table 1.** Correspondent CS values for C-Level flags representation, adapted from (Galkin et al., 2013)

| Confidence Score | C-level |
| --- | --- |
| 81..100 | 11 |
| 61..80 | 22 |
| 41..60 | 33 |
| 21..40 | 44 |
| 0..20 | 55 |

.

**Table 2.** Ionosonde locations and correspondent closest IGS grid data location.

| Ionosonde: Lat, Lon | Closest IGS data Lat, Lon |
|---|---|
| BVJ03: 2.8°, -60.7° | 2.5°, -60.0° |
| CAJ2M: -22.7°, -45.0° | -22.5°, -45.0° |
| CGK21: -20.5°, -55.0° | -20.0°, -55.0° |
| FZA0M: -3.9°, -38.4° | -5.0°, -40.0° |
| SAA0K: -2.6°, -44.2° | -2.5°, -45.0° |