# Peer review of "Ionosonde Total Electron Content Evaluation Using IGS Data"

_Annales Geophysicae, 2019_

## Referee Comment (RC1) · Anonymous Referee #1 · 27 Sep 2019

Report on the paper "Ionosonde Total Electron Content Evaluation Using IGS Data" by Telmo dos Santos Klipp et al.

The paper considers a time window of two years to compare the ITEC (ionospheric total electron content) measured by some ionosondes to vTEC (vertical total electron content) given by IGS maps. The authors say that ITEC is significantly lower than vTEC and uses the adapted α-Chapman analytical representation of the topside proposed by Jakowski (2005) to fill the gap.

My major concern is about the novelty of the work. It is well-known that ITEC is significantly lower than vTEC and it is somewhat expected that introducing for the ionosonde a topside representation extending till 20000 km the gap is reduced.

To increase the scientific content of the paper I invite the authors to compare at least two different topside analytical representations, in order to evaluate which one could be considered the most reliable for the region under study. For instance, the authors might consider the following paper

M. Pezzopane and A. Pignalberi (2019), The ESA Swarm mission to help ionospheric modeling: a new NeQuick topside formulation for mid-latitude regions, *Scientific Reports* 9:12253, doi:10.1038/s41598-019-48440-6

which has been recently published, consider the new analytical topside formulation proposed by the authors and make a performance comparison between this and the one proposed by Jakowski (2005). Even though the paper by Pezzopane and Pignalberi (2019) is focused on mid latitudes, the authors have recently given a presentation at the IRI workshop held in Nicosia (Cyprus) from 9 to 13 September 2019 in which they have shown that the new Nequick topside formulation is really powerful also at low latitudes.

Other issues:

-when talking about the total electron content until 1000-2000 km measured by an ionosonde they usually talk about ITEC (Ionospheric TEC) and not vTEC.

-kilometer has to be written as "km" and not as "Km".

-the following sentence "….and the adjustment in the plasmaspheric basis electron density was based on differences to IGS data" at page 2 is unintelligible, please rearrange.

-at page 2 the authors write: "..to produce a vertical electron density profile (ionogram)." This is incorrect. An ionosonde records an ionogram and, after applying an inversion process on the ionogram trace, a vertical electron density profile is obtained.

-at page 3, concerning the citations made by authors about the evaluation of autoscaling systems, I suggest to cite also the following papers:

Gilbert JD, Smith RW (1988) A comparison between the automatic ionogram scaling system ARTIST and the standard manual method. Radio Sci 23(6):968–974. doi:10.1029/RS023i006p00968

Enell C-F, Kozlovsky A, Turunen T, Ulich T, Valitalo S, Scotto C, Pezzopane M (2016) Comparison between manual scaling and Autoscala automatic scaling applied to Sodankyla¨ Geophysical Observatory ionograms. Geosci Instrum Method Data Syst 5:53–64. doi:10.5194/gi-5-53-2016

Bamford RA, Stamper R, Cander LR (2008) A comparison between the hourly autoscaled and manually scaled characteristics from the Chilton ionosonde from 1996 to 2004. Radio Sci 43(1):RS1001. doi:10.1029/2005RS003401

M. Pezzopane, V. G. Pillat, and P. R. Fagundes (2017), Automatic scaling of critical frequency foF2 from ionograms recorded at Sao Jose dos Campos, Brazil: a comparison between Autoscala and UDIDA tools, Acta Geophysica 65, 173-187, doi:10.1007/s11600-017-0015-z

-at page 3 the right citation for the QUALSCAN algorithm is

McNamara, L. F. (2006), Quality figures and error bars for autoscaled Digisonde vertical incidence ionograms, Radio Sci., 41, RS4011, doi:10.1029/2005RS003440.

and not Galkin et al. (2013).

-at page 6 replace "(with $i$=1,2,...,$n$)" with "($ei$, $i$=1,2,...,$n$)"

-concerning Fig. 4, on the x axis add also the local time.

-concerning Fig. 4, the legend is confused, there are two "11" and two "22", please check.

-at page 6 remove the sentence "Such coherence has been well explained by Klipp et al. (2019). These authors have analyzed the IGS TEC for the equatorial, low and mid latitudes and also for the same period as presented in this work. It was noticed seasonal TEC dependence with maxima during equinoxes for equatorial and low latitude sectors, but modulated by an overlay effect of the solar flux."

-Figure 5 is useless, I invite the authors to remove it.

-at page 8 the authors write "Figure 7b shows the ionosondes peak of plasma frequency (foF2)....." but I cannot understand how these values have been calculated. Have these mean values been calculated by considering data coming from all the ionosondes? Please, clarify. The same issue stand also for the "Maximum altitude" and the "Plasma Frequency at the Maximum Altitude".

-at page 9 the authors talk about 22,000 km but in the next page they talk about 20,000 km. Please, clarify.

-at page 10 the authors write "Different values for the proportionality coefficient K were examined, and Fig. 8 shows the correspondent TEC differences to IGS in terms of total RMSE." Again, I cannot understand how RMSE values have been calculated? Have these mean values been calculated by considering data coming from all the ionosondes? Please, clarify.

-at pages 11-12 the authors write "Yet, we could observe the matching between ionosonde and IGS TEC seems worse during low ionization periods, mainly nearby June solstice,........." but looking at Figure 9 it does not seem that during June solstice the matching is worse.

---

## Referee Comment (RC2) · Anonymous Referee #2 · 16 Oct 2019

Report on the paper "Ionosonde Total Electron Content Evaluation Using IGS Data" by Telmo dos Santos Klipp et al. angeo-2019-131 The manuscript compares the "Ionosonde Total Electron Content, ITEC", derived from groundbased ionogram measurements, with the "International GNSS Service (IGS) vertical-TEC, vTEC" for a low latitude/equatorial region. The authors use two years of ionogram data from a 5-station Digisonde network in Brazil. Avoiding the mistake made by some of the previous analyses, the authors made careful use of the "confidence level" information contained in the Digisonde ionograms to filter out questionable ionogram data. This careful analysis of the difference between ITEC and vTEC focussing on the equatorial ionosphere anomaly (EIA) region should be published if appropriate revisions and corrections can be made. Here are the major concerns. 1. The authors state that "they noticed" that

[Figure]

ITEC systematically underestimates vTEC, and they explain this by claiming that the ITEC profile integration stops at 900 km. Both claims are not quite correct. Firstly, the original ITEC paper by Reinisch and Huang [2001], which the authors have cited, shows that the height integration for the ITEC calculation goes to infinity, and is not stopped at ~900 km. The Digisonde calculations of ITEC assume an ïĄą-Chapman topside profile with constant scale height Hm. Secondly, extensive studies by Belehaki et al. [e.g., 2004, 2012] had shown as early as 2004 that the Digisonde ITEC systematically underestimates vTEC; Belehaki's explanation was that a constant scale height Hm (calculated from the bottomside profile for heights near hmF2) makes the topside profile decay too rapidly with height. They concluded that the plasma above about 900 km is practically not included in the Digisonde's ITEC value. Instead of saying "they noticed" the underestimate, it might be more correct to say that the Belehaki et al. results were "confirmed" to also apply in the equatorial region. 2. Since the authors try providing a comprehensive review of the ITEC technique, why do they not mention the "Vary-Chap topside profile" that was introduced by Reinisch et al. [2007] based on a topside scale height H(h) that varies continuously with height h, see also Nsumei et al. [2012]. 3. What is the meaning of RMSE in eq. (1)? The "error" is defined as the "difference between TEC values". Which TEC values? Is the error defined as the deviation from a mean? The mean over what samples? It would be helpful if the authors would provide a clear description, and explain what is plotted in Figures 6 and 7. 4. The paper makes a clear point in emphasizing that any high-volume data analysis depends on the availability of automatically processed data, and of automatically generated data confidence scores, this is very good and important. The Brazilian Digisondes have used the ARTIST-5 autoscaler (as stated on p3/25), so why is there such lengthy discussion of the performance of ARTIST 4.0, 4.5, and AUTOSCALA when none of these were used for the analysis of the 2016-2017 data reported in this paper? A short note may suffice to alert the reader. (By the way, older Digisonde data can be automatically reprocessed with ARTIST-5 using SAO-Explorer. Have you checked whether AUTOSCALA determines hmF2, which is a required input for the construction of the topside profile in Eq.

2?). 5. Figures 7c and 7d introduce the "Maximum Altitude" and "Plasma Frequency". How is the Maximum Altitude defined?

Some minor concerns: Careful proofreading of the text is required, e.g. gaped echoes traces → gapped echo traces, etc. It would be useful to systematically refer to "ITEC" (as derived from ionograms) and "vTEC" or "IGSTEC" (obtained from IGS maps), or similar notation, which would make it easier for the reader to follow the discussions.
* * *
Belehaki, A., B.W. Reinisch, and N. Jakowski (2004), Plasmaspheric electron content derived from GPS TEC and digisonde ionograms, Adv. Spac. Res., 33, 833-837. Belehaki, A., I. Kutiev, B. Reinisch, N. Jakowski, P. Marinov, I. Galkin, C. Mayer, I. Tsagouri, T. Herekakis (2012), Verification of the TSMP-assisted digisonde topside profiling technique, Acta Geophysica, 04/2012. 432-452, doi:10.2478/s11600-009-0052-3. Reinisch, B.W., P. Nsumei, X. Huang, and D.K. Bilitza, Modeling the F2 topside and plasmasphere for IRI using IMAGE/RPI, and ISIS data, Adv. Space Res., 39, 731-738, 2007. Nsumei, P., B.W. Reinisch, X. Huang, and D. Bilitza (2012), New Vary-Chap profile of the topside ionosphere electron density distribution for use with the IRI Model and the GIRO real time data, Radio Sci., doi:10.1029/2012RS004989.

---

## Referee Comment (RC3) · Anonymous Referee #1 · 16 Oct 2019

The reviewer says: "1. The authors state that "they noticed" that ITEC systematically underestimates vTEC, and they explain this by claiming that the ITEC profile integration stops at 900 km. Both claims are not quite correct. Firstly, the original ITEC paper by Reinisch and Huang [2001], which the authors have cited, shows that the height integration for the ITEC calculation goes to infinity, and is not stopped at 900 km."

This is not true. For the time window (2016-2017) considered by the authors the ITEC (Ionospheric - not Ionosonde - Total Electron Content) given as output by digisondes is the one calculated to approximately 1000 km of altitude.

The reviewer continues to saying: "The Digisonde calculations of ITEC assume an alfa-Chapman topside profile with constant scale height Hm. Secondly, extensive studies by Belehaki et al. [e.g., 2004, 2012] had shown as early as 2004 that the Digisonde

ITEC systematically underestimates vTEC; Belehaki's explanation was that a constant scale height Hm (calculated from the bottomside profile for heights near hmF2) makes the topside profile decay too rapidly with height. They concluded that the plasma above about 900 km is practically not included in the Digisonde's ITEC value. Instead of saying "they noticed" the underestimate, it might be more correct to say that the Belehaki et al. results were "confirmed" to also apply in the equatorial region."

The authors cannot consider what the reviewer is claiming, especially "the Belehaki et al. results were "confirmed" to also apply in the equatorial region" because the situation here is completely different from that faced by Belehaki et al. I repeat, ITEC values considered by the authors for the time window 2016-2017 are those calculated till 1000 km of altitude and not beyond.

---

## Referee Comment (RC4) · Anonymous Referee #2 · 22 Oct 2019

Anonymous Referee #1 The reviewer says: "1. The authors state that "they noticed" that ITEC systematically underestimates vTEC, and they explain this by claiming that the ITEC profile integration stops at 900 km. Both claims are not quite correct. Firstly, the original ITEC paper by Reinisch and Huang [2001], which the authors have cited, shows that the height integration for the ITEC calculation goes to infinity, and is not stopped at 900 km." This is not true. For the time window (2016-2017) considered by the authors the ITEC (Ionospheric - not Ionosonde - Total Electron Content) given as output by digisondes is the one calculated to approximately 1000 km of altitude. Reply by the reviewer: The ionosonde TEC calculation in the Digisonde is performed as part of the NHPC pro-

gram, and Reinisch and Huang [2001] state that the analytic integration for the topside goes from 0→inf. The DIDBase and SAO characteristic #38 contain this TEC value. In the literature this ionosonde-derived TEC value is occasionally referred to as ITEC. It could of course be that the authors have numerically recalculated the topside content up to 900 or 1000 km with the alpha-Chapman profile and constant scale height Hm. But even if they did, it could not explain the observed underestimation by the ionosonde technique. The reason is not an abrupt cutoff at 900 km, but an invalid scale height Hm (of ∼75km) that is way too small a value for heights above ∼700 km as discussed in several papers since 2001. The reviewer continues to saying: "The Digisonde calculations of ITEC assume an alpha-Chapman topside profile with constant scale height Hm. Secondly, extensive studies by Belehaki et al. [e.g., 2004, 2012] had shown as early as 2004 that the Digisonde ITEC systematically underestimates vTEC; Belehaki's explanation was that a constant scale height Hm (calculated from the bottomside profile for heights near hmF2) makes the topside profile decay too rapidly with height. They concluded that the plasma above about 900 km is practically not included in the Digisonde's ITEC value. Instead of saying "they noticed" the underestimate, it might be more correct to say that the Belehaki et al. results were "confirmed" to also apply in the equatorial region." The authors cannot consider what the reviewer is claiming, especially "the Belehaki et al. results were "confirmed" to also apply in the equatorial region" because the situation here is completely different from that faced by Belehaki et al. I repeat, ITEC values considered by the authors for the time window 2016-2017 are those calculated till 1000 km of altitude and not beyond.

Reply by the reviewer: Please see my response above for the 1000 km upper integration limit. You are right, the Belehaki et al. papers are for a completely different situation. This is why I recommended that your paper be published since it applies to the equatorial region.

---

## Short Comment (SC1) · 31 Oct 2019

Reviewer #1 Comments:

Report on the paper "Ionosonde Total Electron Content Evaluation Using IGS Data" by Telmo dos Santos Klipp et al.

The paper considers a time window of two years to compare the ITEC (ionospheric total electron content) measured by some ionosondes to vTEC (vertical total electron content) given by IGS maps. The authors say that ITEC is significantly lower than vTEC and uses the adapted $\alpha$-Chapman analytical representation of the topside proposed by Jakowski (2005) to fill the gap.

My major concern is about the novelty of the work. It is well-known that ITEC is significantly lower than vTEC and it is somewhat expected that introducing for the ionosonde a topside representation extending till 20000 km the gap is reduced.

To increase the scientific content of the paper I invite the authors to compare at least two different topside analytical representations, in order to evaluate which one could be considered the most reliable for the region under study. For instance, the authors might consider the following paper

M. Pezzopane and A. Pignalberi (2019), The ESA Swarm mission to help ionospheric modeling: a new NeQuick topside formulation for mid-latitude regions, Scientific Reports 9:12253, doi:10.1038/s41598-019-48440-6

which has been recently published, consider the new analytical topside formulation proposed by the authors and make a performance comparison between this and the one proposed by Jakowski (2005). Even though the paper by Pezzopane and Pignalberi (2019) is focused on mid latitudes, the authors have recently given a presentation at the IRI workshop held in Nicosia (Cyprus) from 9 to 13 September 2019 in which they have shown that the new Nequick topside formulation is really powerful also at low latitudes.

Authors: We would like to thank the reviewer for the very important comments and suggestions. The modifications in the manuscript (included as supplement material) were included in red color.

We are deeply grateful to the reviewer for bringing us this up-to-date suggestion to substantially improve the novelty of our paper. We have read the suggested paper (and references therein), and also got in touch with the authors (Dr. Pezzopane and Dr. Pignalberi) who provided the 2 two-dimensional grids (numerically) derived from Swarm satellites. Then, we were able to implement this topside reconstruction technique using ionosonde data - the NeQuick topside formulation, based on a semi-Epstein layer with modeled scale height as a function of a corrected version of the Ho empirical parameter, and the manuscript was significantly improved. The section "3 Experiments and
results" was divided into 3 subsections: "3.1 Adapted $\alpha$-Chapman", "3.2 NeQuick top-side formulation", and "3.3 Comparative evaluation". Figures 9 was updated including NeQuick formulation and divided into 2 figures (figures 9 and 10), to better present the results. Abstract and conclusion section were substantially altered.

Other issues: -when talking about the total electron content until 1000-2000 km measured by an ionosonde they usually talk about ITEC (Ionospheric TEC) and not vTEC.

Authors: We adjusted the correspondent references to Ionospheric TEC (ITEC).

-kilometer has to be written as "km" and not as "Km".

Authors: We corrected the manuscript.

-the following sentence "....and the adjustment in the plasmaspheric basis electron density was based on differences to IGS data" at page 2 is unintelligible, please rearrange.

Authors: The sentence was modified to avoid misunderstanding and to reflect the major changes in the manuscript.

-at page 2 the authors write: "..to produce a vertical electron density profile (ionogram)." This is incorrect. An ionosonde records an ionogram and, after applying an inversion process on the ionogram trace, a vertical electron density profile is obtained.

Authors: We changed the manuscript to avoid this inaccurate explanation.

-at page 3, concerning the citations made by authors about the evaluation of autoscaling systems, I suggest to cite also the following papers: Gilbert JD, Smith RW (1988) A comparison between the automatic ionogram scaling system ARTIST and the standard manual method. Radio Sci 23(6):968–974. doi:10.1029/RS023i006p00968 Enell C-F, Kozlovsky A, Turunen T, Ulich T, Valitalo S, Scotto C, Pezzopane M (2016) Comparison between manual scaling and Autoscala automatic scaling applied to Sodankyla ÌL Geophysical Observatory ionograms. Geosci Instrum Method Data Syst 5:53–64. doi:10.5194/gi-5-53-2016 Bamford RA, Stamper R, Cander LR (2008) A comparison

between the hourly autoscaled and manually scaled characteristics from the Chilton ionosonde from 1996 to 2004. Radio Sci 43(1):RS1001. doi:10.1029/2005RS003401 M. Pezzopane, V. G. Pillat, and P. R. Fagundes (2017), Automatic scaling of critical frequency foF2 from ionograms recorded at Sao Jose dos Campos, Brazil: a comparison between Autoscala and UDIDA tools, Acta Geophysica 65, 173-187, doi:10.1007/s11600-017-0015-z

Authors: The suggested references were included in the section "1.2.1 Ionogram quality" subsection, which was improved following also reviewer # 2 suggestions.

-at page 3 the right citation for the QUALSCAN algorithm is McNamara, L. F. (2006), Quality figures and error bars for autoscaled Digisonde vertical incidence ionograms, Radio Sci., 41, RS4011, doi:10.1029/2005RS003440. and not Galkin et al. (2013).

Authors: The suggested citation referencing QualScan algorithm was included to the sentence.

-at page 6 replace "(with i=1,2,...,n)" with "(ei, i=1,2,...,n)"

Authors: We did not find the text "(with i=1,2,...,n)" on page 6. Assuming the reviewer meant to replace "(ei, i=1,2,...,n)", the text was corrected in that way.

-concerning Fig. 4, on the x axis add also the local time.

Authors: We can not simply add local time to x-axis, since the brazilian ionosondes used are located at 2 different time zones (UTC-3 and UTC-4).

-concerning Fig. 4, the legend is confused, there are two "11" and two "22", please check.

Authors: In fact, Fig. 4 shows the number of occurrences of ionograms C-level 11 (blue curve), 22 (green curve), and we also included a red curve to show the number of occurrences of both 11+22 (basically the sum of 11 and 22 curves), which represent the data actually used in this work. We included the comment "see red curve in Fig. 4"

to better clarify this issue.

-at page 6 remove the sentence "Such coherence has been well explained by Klipp et al. (2019). These authors have analyzed the IGS TEC for the equatorial, low and mid latitudes and also for the same period as presented in this work. It was noticed seasonal TEC dependence with maxima during equinoxes for equatorial and low latitude sectors, but modulated by an overlay effect of the solar flux."

Authors: The sentence was removed.

-Figure 5 is useless, I invite the authors to remove it.

Authors: The first paragraph of section 3 "Experiments and results" bring us interesting evaluations of figures 5 and 6, with reasonable explanations about differences in the plots of Figure 5, which strengthen the authors decision of working only with C-level 11 and 22 data in the experiments.

-at page 8 the authors write "Figure 7b shows the ionosondes peak of plasma frequency (foF2)....." but I cannot understand how these values have been calculated. Have these mean values been calculated by considering data coming from all the ionosondes? Please, clarify. The same issue stand also for the "Maximum altitude" and the "Plasma Frequency at the Maximum Altitude".

Authors: Yes, the values presented in Fig. 7a, 7b, 7c and 7d are calculated as the daily mean, considering data in all ionosondes. The text was improved and the caption of Fig 7 was updated to clarify that.

-at page 9 the authors talk about 22,000 km but in the next page they talk about 20,000 km. Please, clarify.

Authors: For different GNSS satellite constellations, the satellites altitudes are not the same but also are not too different. We believe it is appropriate to mention "approximately" 20,000 km as the satellites altitudes, and to use the same 20,000 km for the experiments. The text was corrected that way.

-at page 10 the authors write "Different values for the proportionality coefficient K were examined, and Fig. 8 shows the correspondent TEC differences to IGS in terms of total RMSE." Again, I cannot understand how RMSE values have been calculated? Have these mean values been calculated by considering data coming from all the ionosondes? Please, clarify.

Authors: Yes, the RMSE values presented in Fig. 8 are the global mean for the whole period evaluated, considering all ionosondes. We changed the sentence to better explain this.

-at pages 11-12 the authors write "Yet, we could observe the matching between ionosonde and IGS TEC seems worse during low ionization periods, mainly nearby June solstice,........." but looking at Figure 9 it does not seem that during June solstice the matching is worse.

Authors: The conclusion section was completely reviewed to consider the new results, and the mistake was corrected.

Please also note the supplement to this comment:
https://www.ann-geophys-discuss.net/angeo-2019-131/angeo-2019-131-SC1-supplement.pdf

**Supplement:**

[revised manuscript text omitted]

---

## Short Comment (SC2) · 31 Oct 2019

Reviewer #2 Comments:

Report on the paper "Ionosonde Total Electron Content Evaluation Using IGS Data"by Telmo dos Santos Klipp et al. angeo-2019-131 The manuscript compares the "Ionosonde Total Electron Content, ITEC", derived from ground based ionogram measurements, with the "International GNSS Service (IGS) vertical-TEC, vTEC" for a low latitude/equatorial region. The authors use two years of ionogram data from a 5-station Digisonde network in Brazil. Avoiding the mistake made by some of the previous analyses, the authors made careful use of the "confidence level" information contained in the Digisonde ionograms to filter out questionable ionogram data. This careful analy-

sis of the difference between ITEC and vTEC focussing on the equatorial ionosphere anomaly (EIA) region should be published if appropriate revisions and corrections can be made.

Here are the major concerns. 1. The authors state that "they noticed" that ITEC systematically underestimates vTEC, and they explain this by claiming that the ITEC profile integration stops at 900 km. Both claims are not quite correct. Firstly, the original ITEC paper by Reinisch and Huang [2001], which the authors have cited, shows that the height integration for the ITEC calculation goes to infinity, and is not stopped at âĹij900 km. The Digisonde calculations of ITEC assume an ðİŻij-Chapman topside profile with constant scale height Hm. Secondly, extensive studies by Belehaki et al. [e.g., 2004, 2012] had shown as early as 2004 that the Digisonde ITEC systematically underestimates vTEC; Belehaki's explanation was that a constant scale height Hm (calculated from the bottomside profile for heights near hmF2) makes the topside profile decay too rapidly with height. They concluded that the plasma above about 900km is practically not included in the Digisonde's ITEC value. Instead of saying "they noticed" the underestimate, it might be more correct to say that the Belehaki et al. results were "confirmed" to also apply in the equatorial region.

Authors: We would like to thank the reviewer for providing very interesting and important suggestions to improve the manuscript. The modifications made in manuscript (attached as supplement material) are in blue color.

Considering the discussion where reviewer #1 reply to this reviewer #2 comment: "This is not true. For the time window (2016-2017) considered by the authors the ITEC (Ionospheric - not Ionosonde - Total Electron Content) given as output by digisondes is the one calculated to approximately 1000 km of altitude." Reviewer #2 added: "The ionosonde TEC calculation in the Digisonde is performed as part of the NHPC program, and Reinisch and Huang [2001] state that the analytic integration for the topside goes from 0→inf. The DIDBase and SAO characteristic #38 contain this TEC value. In the literature this ionosonde-derived TEC value is occasionally referred to as ITEC. It

could of course be that the authors have numerically recalculated the topside content up to 900 or 1000 km with the alpha-Chapman profile and constant scale height Hm. But even if they did, it could not explain the observed underestimation by the ionosonde technique. The reason is not an abrupt cutoff at 900 km, but an invalid scale height Hm (of âĹij75km) that is way too small a value for heights above âĹij700 km as discussed in several papers since 2001." And also the reviewer #1 wrote: "The authors cannot consider what the reviewer is claiming, especially "the Belehaki et al. results were "confirmed" to also apply in the equatorial region" because the situation here is completely different from that faced by Belehaki et al. I repeat, ITEC values considered by the authors for the time window 2016-2017 are those calculated till 1000 km of altitude and not beyond." Reviewer #1 added: "Please see my response above for the 1000 km upper integration limit. You are right, the Belehaki et al. papers are for a completely different situation. This is why I recommended that your paper be published since it applies to the equatorial region." We would like to clear that we have used ionosonde TEC provided in SAO files (digisonde outputs), now referenced in manuscript as ITEC, and the differences to IGS vTEC were observed. We reconstructed the topside profile using only alpha-Chapman and constant scale height, and it was observed, even if we extended the maximum altitude, little difference in the final density integration (TEC). It is clear a different modeling approach for the topside profile is necessary to appropriate comparison to IGS vTEC. And we tried 2 different approaches in this work: the adapted alpha-Chapman (which uses a scale for plasmasphere), and NeQuick with variable scale height. Both procedures provided similar results and were able to consider plasmaspheric TEC and reduce RMSE. It was not considered relevant to discuss the well expected differences between ITEC and TEC. We have updated the manuscript, indicating these differences were expected, and emphasized our major contributions.

2. Since the authors try providing a comprehensive review of the ITEC technique, why do they not mention the "Vary-Chap topside profile" that was introduced by Reinisch et al. [2007] based on a topside scale height H(h) that varies continuously with height h,

see also Nsumei et al.[2012].

Authors: We included a brief overview of different topside profiles techniques, and Vary-Chap topside profile references were included.

3. What is the meaning of RMSE in eq. (1)? The "error" is defined as the "difference between TEC values". Which TEC values? Is the error defined as the deviation from a mean? The mean over what samples? It would be helpful if the authors would provide a clear description, and explain what is plotted in Figures 6 and 7.

Authors: RMSE stands for "root mean squared error", as mentioned before equation 1. The "error" is defined as the difference between 2 values, which in our work come from IGS and ionosonde. We tried to improve the text and figure captions to avoid misunderstanding.

4. The paper makes a clear point in emphasizing that any high-volume data analysis depends on the availability of automatically processed data, and of automatically generated data confidence scores, this is very good and important. The Brazilian Digisondes have used the ARTIST-5 autoscaler (as stated on p3/25), so why is there such lengthy discussion of the performance of ARTIST 4.0, 4.5, and AUTOSCALA when none of these were used for the analysis of the 2016-2017 data reported in this paper? A short note may suffice to alert the reader. (By the way, older Digisonde data can be automatically reprocessed with ARTIST-5 using SAO-Explorer. Have you checked whether AUTOSCALA determines hmF2, which is a required input for the construction of the topside profile in Eq. 2?).

Authors: The discussion about autoscaling systems was reduced. Other autoscale systems (e.g. AUTOSCALA) were not used.

5. Figures 7c and 7d introduce the "Maximum Altitude" and "Plasma Frequency". How is the Maximum Altitude defined?

Authors: The maximum altitude shown is the one provided by ionosonde SAO files.

Both, the maximum altitude and the plasma frequency measured at maximum altitude were daily averaged considering all ionosondes. We tried to improve the manuscript to clear this point.

Some minor concerns: Careful proofreading of the text is required, e.g. gaped echoes traces → gapped echo traces, etc. It would be useful to systematically refer to "ITEC" (as derived from ionograms) and "vTEC" or "IGSTEC" (obtained from IGS maps), or similar notation, which would make it easier for the reader to follow the discussions.

Authors: We tried to correct the text, and adopted ITEC and vTEC to differentiate the TEC sources. _______________ Belehaki, A., B.W. Reinisch, and N. Jakowski (2004), Plasmaspheric electron content derived from GPS TEC and digisonde ionograms, Adv. Spac. Res., 33, 833-837. Belehaki, A., I. Kutiev, B. Reinisch, N. Jakowski, P. Marinov, I. Galkin, C. Mayer, I. Tsagouri, T. Herekakis (2012), Verification of the TSMP-assisted digisonde topside pro- filing technique, Acta Geophysica, 04/2012. 432-452, doi:10.2478/s11600-009-0052- 3. Reinisch, B.W., P. Nsumei, X. Huang, and D.K. Bilitza, Modeling the F2 topside and plasmasphere for IRI using IMAGE/RPI, and ISIS data, Adv. Space Res., 39, 731-738, 2007. Nsumei, P., B.W. Reinisch, X. Huang, and D. Bilitza (2012), New Vary-Chap profile of the topside ionosphere electron density distribution for use with the IRI Model and the GIRO real time data, Radio Sci., doi:10.1029/2012RS004989.

Please also note the supplement to this comment:
https://www.ann-geophys-discuss.net/angeo-2019-131/angeo-2019-131-SC2-supplement.pdf

———————————————

[Figure]

**Supplement:**

[revised manuscript text omitted]

---

## Referee Report (RR1)

Report on the revised version paper "Ionosonde Total Electron Content Evaluation Using IGS Data" by Telmo dos Santos Klipp et al.

The authors properly satisfied all my previous requests. The paper now is improved and more significant from a scientific point of view.
Before suggesting it for publication, I would suggest the authors to consider the following minor comments:

-page 1, line 10, replace "…and plasmasphere, and the NeQuick topside formulation." with "…and plasmasphere, and a corrected version of the NeQuick topside formulation."

-page 2, line 8, replace "…optimization, and the NeQuick topside formulation (Pezzopane and Pignalberi, 2019)." with "…optimization, and a corrected version of the NeQuick topside formulation (Pezzopane and Pignalberi, 2019)."

-fig. 10, replace "Km" with "km"

-pag. 11, line 4, delete "Pignalberi et al., 2018", they do not talk about NeQuick there.

-pag 12, line 2, delete "Pignalberi et al., 2018", they do not talk about NeQuick there.

-pag. 12, lines 8-12, replace "In Pignalberi et al. (2018), the authors propose a statistical assessment to create a two-dimensional grid to map $H_0$ values, as a function of f0F2 and hmF2. Pezzopane and Pignalberi (2019) propose a new calculation of $H_0$ using also a combination estimatives, derived from different Swarm satellites datasets." with "Pezzopane and Pignalberi (2019) proposed a new formulation for $H_0$ based on electron density measurements made by the Swarm satellite constellation."

-pag. 12, line 12, replace "…the values of $H_{0,AC}$ and $H_{0,B}$ for a given input ($f_0$F2, $hm$F2). The grids…" with "…the values of $H_{0,AC}$ and $H_{0,B}$ as a function of $f_0$F2 and $hm$F2. Specifically, the grids…"

-pag. 12, line 13, when talking about IRI UP, please, put the following references:

A. Pignalberi, M. Pezzopane, R. Rizzi, and I. Galkin (2018), Effective Solar Indices for Ionospheric Modeling: A Review and a Proposal for a Real-Time Regional IRI, *Surveys in Geophysics* 39:125–167, https://doi.org/10.1007/s10712-017-9438-y.

A. Pignalberi, M. Pietrella, M. Pezzopane, and R. Rizzi 2018, Improvements and validation of the IRI UP method under moderate, strong, and severe geomagnetic storms, *Earth, Planets and Space 70:180*, doi:10.1186/s40623-018-0952-z.

-pag. 15, delete the last sentence "It is relevant to mention that the vertical electron density distribution in the plasmasphere was determined based on a new criterion introduced in this work in which it is applied an adjustment of plasmaspheric basis density, as defined by (Jakowski, 2005), using an optimal factor (K) to minimize the global RMSE." you have already highlighted this concept previously in the text, it is not necessary to reiterate it at the end of the paper.

---

## Author Response (AR2)

**Reviewer #1 Comments:**

Report on the revised version paper "Ionosonde Total Electron Content Evaluation Using IGS Data" by Telmo dos Santos Klipp et al.

The authors properly satisfied all my previous requests. The paper now is improved and more significant from a scientific point of view. Before suggesting it for publication, I would suggest the authors to consider the following minor comments:

- page 1, line 10, replace "…and plasmasphere, and the NeQuick topside formulation." with "…and plasmasphere, and a corrected version of the NeQuick topside formulation."

Authors: The sentence was modified.

- page 2, line 8, replace "…optimization, and the NeQuick topside formulation (Pezzopane and Pignalberi, 2019)." with "…optimization, and a corrected version of the NeQuick topside formulation (Pezzopane and Pignalberi, 2019)."

Authors: The sentence was modified.

- fig. 10, replace "Km" with "km"

Authors: The Fig. 10 does not show "Km", but Fig. 7 does, and it was corrected to "km".

- pag. 11, line 4, delete "Pignalberi et al., 2018", they do not talk about NeQuick there.

Authors: The reference was removed.

- pag 12, line 2, delete "Pignalberi et al., 2018", they do not talk about NeQuick there.

Authors: The reference was removed.

- pag. 12, lines 8-12, replace "In Pignalberi et al. (2018), the authors propose a statistical assessment to create a two-dimensional grid to map H0 values, as a function of f0F2 and hmF2. Pezzopane and Pignalberi (2019) propose a new calculation of H0 using also a combination estimatives, derived from different Swarm satellites datasets." with "Pezzopane and Pignalberi (2019) proposed a new formulation for H0 based on electron density measurements made by the Swarm satellite constellation."

Authors: The sentence was modified.

- pag. 12, line 12, replace "…the values of H0,AC and H0,B for a given input (f0F2, hmF2). The grids…" with "…the values of H0,AC and H0,B as a function of f0F2 and hmF2. Specifically, the

Grids…"

Authors: The sentence was modified.

- pag. 12, line 13, when talking about IRI UP, please, put the following references:

A. Pignalberi, M. Pezzopane, R. Rizzi, and I. Galkin (2018), Effective Solar Indices for Ionospheric Modeling: A Review and a Proposal for a Real-Time Regional IRI, Surveys in Geophysics 39:125–167, https://doi.org/10.1007/s10712-017-9438-y.

A. Pignalberi, M. Pietrella, M. Pezzopane, and R. Rizzi 2018, Improvements and validation of the IRI UP method under moderate, strong, and severe geomagnetic storms, Earth, Planets and Space 70:180, doi:10.1186/s40623- 018-0952-z.

Authors: The references were included in the sentence.

- pag. 15, delete the last sentence "It is relevant to mention that the vertical electron density distribution in the plasmasphere was determined based on a new criterion introduced in this work in which it is applied an adjustment of plasmaspheric basis density, as defined by (Jakowski, 2005), using an optimal factor (K) to minimize the global RMSE." you have already highlighted this concept previously in the text, it is not necessary to reiterate it at the end of the paper.

Authors: The sentence was removed.

**Reviewer #2 Comments:**

- Grammar and style of the text should be edited. For example: "... decay too rapidly over 800-900 km" should say: ... decay too rapidly above ~800 km altitude

Authors: The sentences were edited as suggested.